# Rat Model of Quadriceps Contracture by Joint Immobilization

**DOI:** 10.3390/biology11121781

**Published:** 2022-12-07

**Authors:** Kanokwan Suwankanit, Miki Shimizu

**Affiliations:** 1Department of Veterinary Diagnostic Imaging, Faculty of Agriculture, Tokyo University of Agriculture and Technology, 3-5-8 Saiwai-cho, Tokyo 183-0054, Japan; 2Department of Clinical Sciences and Public Health, Faculty of Veterinary Science, Mahidol University, Nakhon Pathom 73170, Thailand

**Keywords:** joint range of motion, muscle weight, muscle length, cross-sectional area of muscle fiber, collagen percentage, quadriceps contracture

## Abstract

**Simple Summary:**

Muscle contracture can occur in the quadriceps muscle. At the time of diagnosis, the muscle contracture condition is typically advanced, and treatment prognosis is poor. Therefore, to investigate the pathophysiology, early diagnostic method, treatment efficacy, and physical therapy efficacy in future research, we evaluated an easily performed animal model that can allow prolonged knee joint immobilization with minimal trauma to the experimental animals. We created a contracture model using the quadriceps muscle, a relatively large muscle that generates muscle contracture and facilitates future evaluation and approach. We compared previously reported methods of modeling muscle contracture: casts, Velcro hook-and-loop fasteners, and steel wires. After two weeks of immobilization, the range of joint motion, muscle weight, and histological changes in muscle fiber were evaluated. The results showed that the spiral steel wire method was the superior immobilization method.

**Abstract:**

Muscle contracture is an abnormal pathologic process resulting in fibrosis and muscle atrophy, which can lead to limitation of joint motion. To establish a diagnostic method to detect muscle contracture and a method to control its progression, we investigated an appropriate method to create an animal model of quadriceps contracture using rats. Eighteen Wistar rats were divided into three groups, and bilateral hindlimbs were immobilized with either a cast (Group I), a Velcro hook-and-loop fastener (Group V), or steel wire (Group S) with the knee and ankle joints in extension position for two weeks. Five rats in a control group (Group C) were not immobilized. After two weeks, the progression of quadriceps contracture was assessed by measuring the range of joint motion and pathohistological changes. Muscle atrophy and fibrosis were observed in all immobilization groups. The knee joint range of motion, quadriceps muscle weight, and muscle fiber size decreased only in Group S compared to the other immobilization groups. Stress on rats due to immobilization was less in Group S. These results indicate that Group S is the superior quadriceps contracture model. This model aids research investigating diagnostic and therapeutic methods for muscle contracture in humans and animals.

## 1. Introduction

Skeletal muscle contracture is defined as an abnormal shortening of the tendon–muscle unit [1,2,3], characterized by muscle atrophy and fibrosis [1,4]. Muscle contracture can occur in various muscles and has been reported in the quadriceps, deltoid, triceps brachii, and gluteus muscle [5,6,7,8,9]. Quadriceps contracture is encountered as a congenital and acquired disease [4,5,10]. One of the obvious clinical signs of quadriceps contracture is a limited range of knee joint motion due to decreased muscle extensibility [4,6,11], which means a decrease the ability to stretch a muscle to its relaxed length [12]. In experimental studies, two weeks of joint immobilization caused myogenic contracture and limited range of motion of the joint [13]. Myogenic contracture occurs the early stage of joint contracture caused by muscles, fascia, and tendons [14,15]. Other findings include muscle atrophy and fibrosis [4,10,11,16,17,18]. Muscle fibrosis leads to a decrease in muscle elasticity [18], which lowers the muscle’s ability to return to its original length after being stretched [12]. The main pathological findings of muscle atrophy are a decrease in muscle size and cross-sectional area of muscle fiber [18].

Muscle contracture occurs when muscles are held in a shortened position [19,20], and this model can be developed from the muscle atrophy model. For models of hindlimb muscle atrophy, splinting [21], plaster cast [22,23,24], hook-and-loop fastener [25], and spiral wire techniques [26] have been reported. In dogs, splints have been applied to the knee joint under extension to induce muscle atrophy in the biceps femoris, vastus lateralis, and gastrocnemius muscles [21]. In calves, plaster casts have been applied to the knee joint under extension and the ankle joint in full plantar flexion to induce muscle atrophy [24]. In mice, the hook-and-loop fastener (Velcro^®^) method induced tibialis anterior, gastrocnemius, and soleus muscles atrophy equivalent to the cast method, but it was faster and less complicated to apply [25]. The spiral wire method is a technique to immobilize the hindlimb using bonsai steel wire. The soleus muscle weight at 10 days after bilateral immobilized rats by spiral wire technique decreased significantly to 56% compared with the non-immobilized rats [26].

This study aimed to create a quadriceps contracture model that is less stressful to the experimental animals and can be easily implemented from the models of muscle atrophy in previous studies. Due to the large size of the quadriceps muscle, this contracture model can be used to assess the relationship between pathological changes and muscle and joint function, as well as to evaluate methods for diagnosis, prevention, and physical therapy of muscle contractures. The prognosis in humans and animals is positive when treatment is done early [6,27]. Although there are many reports of contracture models of the knee joint [13,15,28,29,30], there are no experimental models designed to induce quadriceps contracture. Therefore, we examined three methods of inducing contracture of the quadriceps muscle: cast, hook-and-loop fastener, and spiral wire. After immobilizing both hindlimbs of rats for two weeks, the joint range of motion, muscle weight and size, muscle fiber size, and collagen percentage in the quadriceps muscle were measured to assess quadriceps contracture. In addition, the treatment time, frequency of rewinding, and condition of the rats were compared for each immobilization method. We hypothesized that the spiral wire method would be easier to apply, have fewer complications, and induce quadriceps contracture than the cast and hook-and-loop fastener methods.

## 2. Materials and Methods

### 2.1. Animals

Twenty-three 8-week-old male SPF Wistar rats (body weight 213.10 ± 7.91 g) were delivered from Nomura Siam International Cooperation. Animals were housed in individually ventilated cages (1 rat per cage) with a 12 h:12 h light–dark cycle. The temperature and humidity were maintained at 24 ± 2 °C and 58%, respectively. Food and water were available ad libitum. Animal care and experimental procedures were done following the Guidelines for the Care and Use of Laboratory Animals of Mahidol University with the approval of the Institute Animal Care and Use Committee (MUVS-2021-05-22). Rats were divided randomly into four groups: the control group (Group C, *n* = 5), cast immobilization group (Group I, *n* = 6), hook-and-loop fastener (Velcro^®^) immobilization group (Group V, *n* = 6), and the spiral wire immobilization group (Group S, *n* = 6).

### 2.2. Immobilization Procedures

Rats in Group C did not receive the immobilization procedure. Rats in Groups I, V, and S were anesthetized with isoflurane, inducted by open drop technique, and maintained using the non-rebreathing circuit of an anesthetic machine. They were maintained in a surgical stage of anesthesia by using isoflurane 1–2% with an oxygen flow rate of 300–600 mL/min. Under anesthesia, the muscles were fully relaxed and exhibited no reflexes, and the rats maintained a stable respiratory rate (50–100 bpm). The hair at both hindlimbs of rats in three immobilization groups was removed using clippers. In all immobilization groups, the knee joint was immobilized in extension, the ankle joint in plantar flexion, and the hip joint was not immobilized. Before applying the cast, Velcro, and spiral wire, bandage displacement and dermatitis prevention procedures were performed using adhesive tape and non-elastic bandage tape (Figure 1A, B) [31]. Four 1.5 × 4 cm pieces of adhesive tape (Fixomull^®^ stretch), and two 1.9 × 28 cm pieces of non-elastic bandage tape (MultiporeTM Sports White Athletic Tape; 3M Japan) were prepared. The rats were placed in a lateral position. Two pieces of adhesive tape per hindlimb—one piece at both the cranial and caudal sides of each hindlimb—were attached at the toe, and the other ends of the tape were attached to a chopstick or 3 mL syringe (Figure 1A). Then, the hindlimb was wrapped with the non-elastic bandage tape with the knee joint in an extended position and the ankle joint in full plantar flexed position (Figure 1B). The contralateral limb underwent the same procedure. After prevention procedures, each immobilization was performed.

In Group I, two 2 × 40 cm casts (ScotchcastTM Plus; 3M Japan) were prepared for both hindlimbs. The cast was wrapped from the toe to the proximal portion of the femur, taking care not to overtighten the cast (Figure 1C). In Group V, two 1.27 × 40 cm hook-and-loop fasteners (Velcro^®^ tape; one-touch belt; 3M Japan) were prepared for both hindlimbs. The Velcro was wound from the toe until the proximal portion of the femur (Figure 1D). In Group S, one 55 cm bonsai steel wire (diameter 2.5 mm) was prepared for both hindlimbs. The rats were placed in a supine position with hip extension. The center of the bonsai steel wire was placed on the dorsal aspect of the L4–L5 spine level and coiled until the toe of both hindlimbs (Figure 1E). After each fixation material was applied, the adhesion tape attached to the chopsticks or syringe was peeled off from the chopstick and attached to the fixation material. The toe ends were not bandaged, so adverse events such as edema or impaired blood flow could be identified. Finally, the fixture was checked for looseness. Then, a vest collar modified from Jang et al. (2016) [32] was placed to protect the bandage. The bandage application time, frequency of bandage rewinding, appearance of the rats, and any adverse events were recorded. The bandage application time was the time taken to apply the bandage to both the right and left sides. Rats were monitored once a day for loosening and any adverse events. Bandages were rewrapped if they became loose or showed any signs of adverse injuries, such as edema, impaired blood flow, skin injury, or necrosis. The frequency of bandage rewinding was defined as the number of times the bandage was changed for each rat. The day of the immobilization procedure was considered Day 1. Bilateral hindlimbs were immobilized for 14 days until Day 14.

### 2.3. Body Weight and Food Intake

Body weight was measured at 10.00 am on Days 1, 2, 5, 8, 11, 14, and 15. Food intake was measured at 10.00 am on Days 2, 5, 8, 11, and 14.

### 2.4. Measuring Joint Angles and Range of Motion

After immobilization for 14 days, on Day 15, the angles of flexion and extension of the hip, knee, and ankle joints of all the rats’ hindlimbs were measured under anesthesia. Rats were anesthetized by open drop technique with isoflurane, and the anesthesia level was maintained with 1–2% isoflurane at an oxygen flow rate of 300–600 mL/min using a rodent face mask. The angle of the joints was evaluated using a goniometer following Millis et al. (2013) [33]. The hip joint angle (0–180°) was measured along the line joining the lateral femoral epicondyle and greater trochanter of the femur and a line joining the tuber sacral and iliac crest when the hip was passively flexed and extended, respectively. The knee joint angle was measured between the long axis of the tibial shaft and the line joining the lateral epicondyle of the femur and the greater trochanter when the knee joint was passively flexed and extended, respectively. Ankle joint angle was measured along the long axis of metatarsal bones III and IV and the long axis of the tibial shaft when the ankle joint was passively flexed and extended, respectively. The range of motion of each joint was calculated by subtracting the angle of the flexed position from that of the extended position.

### 2.5. Muscle Collection and Measurement

After data was recorded, on Day 15 the rats were euthanized, using 30% isoflurane using the open drop chamber technique. They were exposed to isoflurane continuously until one minute after they stopped breathing. After that, bilateral thoracotomy was carried out to ensure death and prevent the rats’ revival [34]. Then, the quadriceps muscles of both sides were harvested and measured for wet weight and muscle size (length × width × height) with a digital vernier caliper. For muscle mass evaluation, the wet weight of the quadriceps muscle in each rat was divided by its body weight (mg/g) [35].

### 2.6. Histopathological Examination

After measurements, the muscles were cut at the muscle belly and fixed in 4% paraformaldehyde for 36–48 h at room temperature. Then, the specimens were fixed in 70% alcohol, dehydrated in ascending concentrations of ethanol up to absolute ethanol, cleared in xylene, and embedded in paraffin. After that, the muscle belly was cut with a microtome of 3 μm to prepare serial transverse tissue sections. The histological slides were deparaffinized in xylene, rehydrated with a graded series of ethanol, and washed in running water. Then, they were stained with hematoxylin and eosin (H&E) [36], Masson’s trichrome [36], and Picrosirius red [37], according to the routine method. H&E stained histological slides were used to assess the minimal Feret’s diameter, diameter, perimeter, and cross-sectional area of muscle fiber. Masson’s trichrome and Picrosirius red stained histological slides were used to measure collagen percentage. All slides were scanned with whole slide digitization (Pannoramic Scan, 3DHISTECH, Budapest, Hungary). ImageJ software (National Institutes of Health, Bethesda, MD, USA) was used for the measurement. To measure muscle fiber minimal Feret’s diameter, diameter, perimeter, and cross-sectional area, 200 round-shaped muscle fibers were randomly selected per quadriceps muscle of each rat [38,39,40,41,42] and measured at ×400 magnification. The mean of 200 muscle fiber measurements was used for statistics. The percentage of collagen to total tissue cross-sectional area, including muscle, perivascular, and interstitial areas, was calculated at ×400 magnification. Areas of muscle overlap or poor staining were excluded. All analyses were performed under conditions where the slides’ group was unknown.

### 2.7. Statistical Analysis

The statistical analyses were performed using GraphPad Prism Version 8 (GraphPad Software Inc., San Diego, CA, USA). Data following a normal distribution are presented as mean ± standard deviation (SD). Comparisons between groups were assessed by one-way analysis of variance (ANOVA), followed by Dunn’s multiple comparisons test. Body weight and food intake were analyzed using two-way ANOVA, followed by Tukey’s multiple comparisons test. Data not following a normal distribution are presented as medians and interquartile ranges, and between-group comparisons were performed using Dunn’s multiple comparisons test. The level of significance was set at *p* < 0.05.

## 3. Results

### 3.1. Bodyweight

The body weight of each group from Day 1 to Day 15 is shown in Table 1. In Group C, the body weight increased with each measurement from Day 2 to Day 14. In the immobilization groups, the body weight decreased after Day 5 compared to that of Group C on the same day. In Group I, the body weight had not increased on any measurement day. In Group V, the body weight decreased on Day 2 compared to Day 1, but no increase in body weight was noted on other measurement days. On the other hand, in Group S, the body weight on Day 2 increased compared to that on Day 1, and the weight after Day 11 increased compared to that before Day 5.

### 3.2. Food Intake

The food intake of each group from Day 2 to Day 14 is represented in Table 2. There was no difference in the food intake between all groups on Day 2. In Group C, the food intake on Day 14 was significantly higher than on Day 2. In Group I, the food intake on Day 5 and Day 11 was lower than Group C on the same day. However, the food intake on Day 11 and 14 was higher than Day 2. In Group V, the food intake on Days 5, 8, and 14 was lower than Group C on the same day, and the food intake did not change during the measurement period. In Group S, the food intake on Day 5 was lower than Group C on the same day. However, the food intake on Day 14 was higher than on Day 2.

### 3.3. Bandage Application Time, the Number of Rewindings, and Adverse Effects

Data on bandage application time is shown in Table 3. The bandage procedures in Groups I and V were simple and did not require special skills. In Group S, the wires were stiff and had to be carefully applied to avoid compression of the inguinal region and hindlimb muscles to avoid hypoxia in the hindlimb. The bandage application time in Group S was the longest, 2.7 and 1.7 times longer than that of Group I and Group V, respectively.

All rats in the immobilized group could move their hip joints and walk; however, the rats in Group S had to move bilateral hindlimbs simultaneously when walking. The number of bandages rewinding was the highest in Group V, 2.5 and 1.7 times higher than that of Group I and Group S, respectively. In Group V, all rats had their bandages rewound at least once, with a maximum of five times per rat. The most common cause of bandage rewinding in Group V was the rats gnawing and removing the bandages after destroying part of the collar. In addition, because Velcro is composed of a combination of plastic and woven fabrics, urine stains and bandage loosening occurred. An ammonia odor due to urine stains also occurred after immobilization for more than two days. In Group I, one rat did not require its bandage rewinding, while the others had theirs rewound once. Group I took the shortest bandaging time; however, the cast was difficult to remove, requiring a cast cutter, taking the longest time to remove. In Group S, one rat did not need its bandage rewinding, but the others had theirs rewound once or twice per rat. Though the destruction of the cast and wire by the rats did not occur, the cause of changing the bandages in Groups I and S was the bandages loosening due to muscle atrophy. During immobilization, no bandages slipped off the toe, as adhesive tape prevented slippage. In Groups I and V, porphyrin secretion—reddish secretion and forms a dark red crust located at the periorbital area—was observed, especially up to Day 7 after bandage application, which indicated conditions were stressful for the rats. In the immobilized group, paraphimosis—which can result from the stress-induced reduction of rats’ grooming behavior in the preputial area—was observed after Day 10 in five rats in Group I, six rats in Group V, and two rats in Group S. They were able to urinate. Rats with paraphimosis lesions were treated using warm normal saline solution, cleaning their penis and preputial area to remove debris. Then, gauze was used to dry the area, and aescin diethylamine salicylate (Reparil-N Gel^®^) was applied to reduce penis swelling. After that, the penis was gently reinserted into the preputial cavity. However, at the end of the experiment, five rats in Groups I and V still had paraphimosis. A skin injury from self-mutilation was seen in one rat in Group V on Day 11. The rat was treated with saline, lidocaine cream, and betadine cream, and the wound had healed completely by Day 14. No other adverse events were found in any of the rats.

### 3.4. Joint Angle and Range of Motion

The angular measurements of the hip, knee, and ankle joints in maximum extension and maximum flexion position on Day 15 are shown in Table 4. The range of motion of each joint, subtracting the angle of the flexion from extension, is represented in Figure 2.

The flexion angle of the ankle and knee joint was significantly higher in all immobilization groups compared to that of Group C. In Group S, the flexion angle of the ankle and knee joint was significantly higher than in Groups I and V. Except for in Group V, the extension angle of the ankle and knee joint was significantly higher in the immobilization groups compared to Group C, and there was no difference between Group I and S. In the hip joint, the flexion angle of Group S was significantly higher than Groups C and V. The range of motion of the ankle joint was decreased in all immobilization groups compared to Group C (*p* = 0.0257 in Group I, *p* = 0.0034 in Group V, and *p* < 0.0001 in Group S). Furthermore, in Group S, the range of motion of the ankle joint was lower than in Group I (*p* = 0.0124). In Group S, the range of motion of the hip and knee joints were lower than to Group C (*p* = 0.0109 and *p* < 0.0001, respectively) and the other immobilization groups (*p* = 0.0442 compared to Group I, *p* = 0.0085 compared to Group V at the hip joint and *p* = 0.0136 compared to Group I, *p* = 0.0328 compared to Group V at the knee joint).

### 3.5. Muscle Weight and Measurement of Quadriceps

Measurements of the weight, width, length, and height of the quadriceps muscle of rats on Day 15 are represented in Table 5. In Group S, the quadriceps muscle weight was lower than Group C (*p* < 0.0001) and the other immobilized groups (*p* = 0.0316 and *p* = 0.0015 compared to Group I and V, respectively). The quadriceps muscle weight standardized by body weight (mg/g) was also significantly lower in Group S compared to that of Group C (*p* < 0.0001) and the other immobilized groups (*p* = 0.0141 and *p* = 0.0027 compared to Group I and V, respectively). The width and height of the quadriceps muscle were lower in all immobilized groups compared to Group C. In Group S alone, the length of the quadriceps muscle was lower than that of Group C (*p* = 0.0027).

### 3.6. Morphological Evaluation of Quadriceps Muscle Fiber

Morphological findings of the quadriceps muscle stained with H&E on Day 15 are represented in Figure 3. A cross-section of skeletal muscle in all groups was arranged in a normal microanatomical arrangement. In all three immobilization groups, the muscle bundle was separated as muscle fascicles surrounded by finely connective tissue perimysium and endomysium. The skeletal muscle fiber showed normal histological architecture with minimal muscle degeneration. Neither cell degeneration nor muscular necrosis was found, though infiltration of fibroblast and connective tissue was found.

The muscles in the three immobilization groups showed atrophic changes in the muscle fibers. The results of the analysis of the minimal Feret’s diameter, diameter, perimeter, and cross-sectional of the quadriceps muscle fibers are shown in Figure 4. The minimal Feret’s diameter of the quadriceps muscle fiber was 56.5 ± 1.1 µm (mean and standard deviation) in Group C, 37.2 ± 1.4 µm in Group I, 34.1 ± 1.6 µm in Group V, and 30.5 ± 1.4 µm in Group S (Figure 4A). The diameter of the quadriceps muscle fiber was 62.9 ± 1.0 µm in Group C, 41.9 ± 1.5 µm in Group I, 38.5 ± 2.0 µm in Group V, and 34.7 ± 1.8 µm in Group S (Figure 4B). The perimeter of the quadriceps muscle fiber was 214.8 ± 3.0 µm in Group C, 141.2 ± 6.2 µm in Group I, 128.6 ± 7.2 µm in Group V, and 116.5 ± 6.4 µm in Group S (Figure 4C). The cross-sectional area of the quadriceps muscle fiber was 3124 ± 104 µm^2^ in Group C, 1390 ± 104 µm^2^ in Group I, 1178 ± 122 µm^2^ in Group V, and 956 ± 95 µm^2^ in Group S (Figure 4D). The minimal Feret’s diameter, diameter, perimeter, and cross-sectional area of the quadriceps muscle in all immobilized groups were lower than Group C. Furthermore, those values were the lowest in Groups S.

### 3.7. Assessment of Quadriceps Muscle Fibrosis

The morphological findings of the quadriceps muscle on Day 15 are represented in Figure 5 and Figure 6, respectively. Collagen fibers are stained blue using Masson’s trichrome staining and the dark-red color by Picrosirius red staining. Cross-sections of skeletal muscle in all three fixation groups showed a normal anatomical arrangement of muscle bundle, with finely positive collagen staining in the endomysium and perimysium area, markedly more prominent than in Group C. The collagen percentage of the quadriceps muscle measured from the stained specimens is represented in Table 6. The collagen percentage in the quadriceps muscle increased in all immobilized groups for both staining methods. No difference in the collagen percentage was observed among the immobilized groups.

## 4. Discussion

We evaluated three immobilization methods to select a rat model of quadriceps contracture, all of which induced atrophy and fibrosis of the quadriceps muscle. The spiral wire method showed a decreased range of motion of knee joints, quadriceps muscle weight, muscle length, and muscle atrophy. Furthermore, the spiral wire method caused less stress to the rats due to immobilization. These results indicate that the spiral wire method is superior to the quadriceps contracture model.

There are different types of materials used to immobilize the hindlimb, with a cast being a common immobilization material. However, prolonged cast immobilization induces hyperalgesia and hindlimb injury [43] and increases the expression of inflammatory mediators and edema of the hindlimb [44]. Aihara et al. (2017) [25] investigated an immobilization model using Velcro material. The Velcro method is simpler than the cast method, with this cheap material available commercially. However, Velcro is made from plastic and fabric, which is easier for the rats to destroy and escape their bandages; rats have a chewing habit and may gnaw on bandages when they feel uncomfortable. In this study, rats in Group V broke part of their collars, and gnawed and removed the bandages. Wire is a commonly available, inexpensive immobilization material that can resist rat gnawing. Onda et al. (2016) [26] reported that wire can immobilize rats’ hindlimbs. Additionally, the spiral wire method does not require special devices. This study examined three materials used in previous studies—cast, Velcro, and wire—to determine which method is more useful in creating quadriceps contracture.

The quadriceps muscle is divided into four muscle heads: the vastus lateralis, vastus intermedius, vastus medialis, and rectus femoris muscles. In rats, the origin of the vastus muscle group is at the proximal part of the femur, and the origin of the rectus femoris muscle arises from the pubis. The four muscles fuse and insert into the upper anterior part of the tibia. The action of these muscles is mainly to extend the knee joint, but the rectus femoris muscle acts as a weak hip joint flexor [45]. Prolonged immobilization of the knee joint in the extended position induces quadriceps contracture [4]. In this study, the knee and ankle joints were immobilized, but the hip joint was not immobilized. This is because the hip joint does not affect the length of the quadriceps muscle and is not necessary to induce quadriceps muscle contracture. Cavalcante et al. (2021) [46] assessed the effect of different hip and knee joint angles on quadriceps muscle length. They found that at 60° and 20° knee flexion, the quadriceps muscle length did not markedly change at different hip angles. This is because the attachment site of the rectus femoris muscle on the ilium does not displace much when the hip angle changes [47]. However, the range of motion of the hip joint was lower in Group S compared to Group C and other immobilization groups. The rats in Group S could walk, but because a single wire was wrapped around both hindlimbs, any movement of the hindlimbs required simultaneous bilateral movement. This may have limited hip motion slightly more in Group S than in the other groups. The ankle joint does not influence quadricep muscle length, because it does not relate to the origin and insertion of the quadriceps [45]. However, it has the beneficial effect of avoiding bandage displacement. Nevertheless, to reduce stress on experimental animals, future studies should examine whether bandages can be maintained, and quadriceps contracture induced without ankle joint immobilization. The range of motion of the knee joint was only lower in Group S compared to that of Group C and other immobilization groups. Thus, it may promote knee joint contracture by inhibiting hip motion, and wires may be stronger than other immobilizing materials.

Muscle contracture is defined as a shortening of tendon–muscle unit accompanied by muscle atrophy and an increase in the accumulation of collagen [1,2,3,4], which precede joint contractures [13,48] As a result of decreased quadriceps length and muscle fiber size, the range of motion of knee joint is decreased [4,27]. In all immobilization groups, quadriceps muscle fiber size decreased and the collagen percentage in muscle tissue increased. The knee joint angle in extension increased in Groups I and S, and angle in flexion increased in all immobilization groups. These results indicated that quadriceps muscle contracture was induced in all immobilization groups. However, only Group S showed a decrease in quadriceps weight, length, and knee joint range of motion compared to that in Group C, and the quadriceps weight, muscle fiber size, and knee joint range of motion decreased compared to the other immobilization groups. These findings indicated that Group S had the most advanced muscle contracture and the pathology had progressed to knee joint contracture among the other immobilization groups. In addition to muscles, joint capsules, tendons, ligaments, cartilage, skin, and bone are involved in loss of a joint’s range of motion [49]. Myogenic contractures involving muscle, fascia, and tendon produced muscle atrophy and are the main etiology of a limited range of joint motion when a joint is immobilized for two weeks [13,15,29,48]. Arthrogenic contractures, due to articular cartilage and periarticular structures, occur after four weeks [13]. Therefore, it was suggested that the cause of the reduced knee joint range of motion in Group S was the shortening of the quadriceps muscle associated with muscle fiber atrophy. Muscle atrophy is defined as a decrease in muscle size due to the loss of cellular substances [50]. Muscle weight and morphological measurements can provide information to detect muscle atrophy. Muscle fiber size measurements can be assessed in terms of minimum Feret’s diameter, diameter, perimeter, or cross-sectional area [51]. The reduction of the muscle diameter, perimeter, and cross-sectional area is the hallmark histological characteristic of muscle atrophy [52,53,54]. Overexpression and accumulation of collagen in the connective tissue of the muscle, including the perimysium, endomysium, and epithelium, are the main characteristics of muscle fibrosis, which decreases skeletal muscle extensibility [18]. Collagen is the main structure of intramuscular connective tissue, providing muscle strength and elasticity [55]. Currently, there are more than 19 types of collagens. Collagen types I and III are found in the intramuscular connective tissue and are responsible for immobilization-induced muscle fibrosis [56]. Type I and III collagen increase in the early stage of immobilization, and Type I collagen increases in the late stage [57]. Type III collagen did not increase its ratio to Type I collagen, induced by three weeks of ankle joint immobilization. It has been noted that Type III collagen expression is independent of the progression of muscle contracture [58]. In this study, two staining methods, Masson’s trichrome stain and Picrosirius red stain, were used to detect muscle fibrosis. Masson’s trichrome stain detects Type I collagen, while Picrosirius red stain detects Types I and III [56]. Polarized light microscopy can distinguish Type I and III collagen in Picrosirius red-stained specimens. In this study, the collagen percentage increased in all immobilized groups for both staining methods. No difference in collagen percentage was observed among the immobilization groups. In this study, to confirm the pathogenesis of muscle contracture, we double-checked the detection of collagen using two staining methods, but did not discriminate the types of collagens.

Prolonged immobilization induces stress [59]. Stress induces a decrease in food intake (anorexia), weight loss, and changes in behavior and appearance in laboratory animals [60,61]. Decreased food intake and body weight are attributed to changes in neurotransmitters and hormones such as corticotropin-releasing hormone (CRH) in response to stress [62,63] and muscle atrophy from muscle disuse [23,39,64]. Onda et al. (2016) [26] reported that using the spiral wire method for six days did not change the food intake, while the body weight decreased compared to the control group. In this study, the food intake and body weight decreased in all immobilization groups compared to that of Group C. However, the changes in Group S tended to be less severe than Groups I and V. Paraphimosis occurs due to decreased grooming behavior caused by pain [65]. Chromodacryorrhoea is associated with pain [66] or environmental stress [67] and is caused by porphyrin secreted by the periorbital Harderian glands [68], which results in ocular fat and red tears [45]. In this study, paraphimosis occurred in all immobilization groups. However, in Group S, the symptoms disappeared with appropriate treatment, while in Groups I and V, the symptoms persisted until the end of the experiment. No chromodacryorrhea occurred in Group S. These findings indicate that the spiral wire method is less stressful to experimental animals than other immobilization methods.

This study has several limitations. First, we focused on muscle changes in myogenic contractures of the quadriceps muscle. However, tendons and fascia are also affected by myogenic contracture [28,30,49]. Second, the reasons for the more advanced muscle atrophy and knee joint contracture in Group S are unclear. Next, functional evaluation using muscle elastic modulus and electromyography will be performed to clarify the mechanisms of muscle atrophy and knee joint contracture. Third, the wire strength was not measured, and the joint angle measurement was subjective. Fourth, this study used male rats in accordance with previously reported muscle atrophy models. Since estrogen and progesterone are involved in the maintenance and increase of muscle mass [69,70], it is unclear how changes in sex hormones during the estrous cycle affect the formation of muscle contractures when female rats are used. In the next study, using this model, we would like to clarify whether muscle fibrosis or muscle elasticity is involved in the progression of the pathology of muscle contracture, as well as early diagnostic methods and effective physical therapy for muscle contracture.

## 5. Conclusions

Based on the results, the spiral wire immobilization method is useful as a model of quadriceps muscle contracture because the wire is available, and rats do not easily remove the fixings. Additionally, the stress during placement is low and does not affect their lives. It causes contracture of the quadriceps muscle and knee joint and muscle atrophy. This model is expected to contribute to future research on the evaluation of pathological mechanisms, diagnosis, and therapeutic methods of muscle contractures in humans and animals.

## Figures and Tables

**Figure 1 biology-11-01781-f001:**
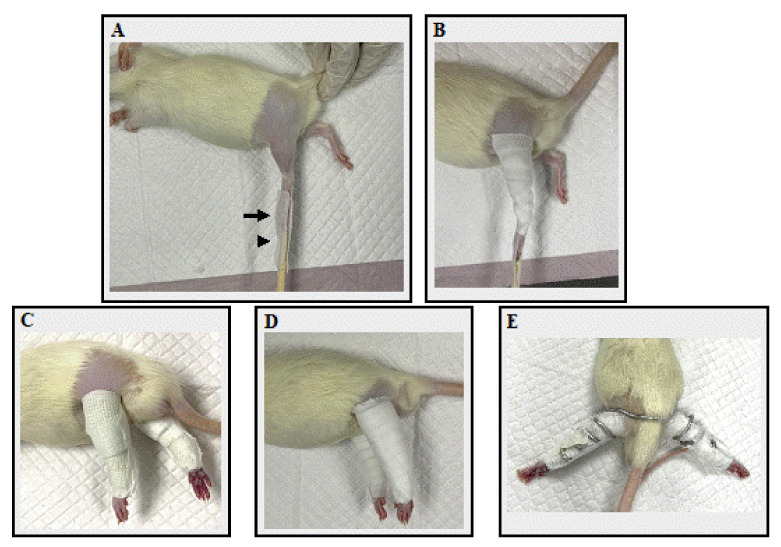
The fixation application processes. (**A**) The application of adhesive tape. The black arrow indicates the rat’s toe, which was attached with adhesive tape, while the black arrowhead indicates a chopstick splint, also attached with adhesive tape. (**B**) Non-elastic bandage tape was applied. (**C**) A rats’ hindlimb in Group I after a cast bandage was applied. (**D**) A rats’ hindlimb in Group V after a Velcro bandage was applied. (**E**) A rats’ hindlimb in Group S after spiral wire was applied.

**Figure 2 biology-11-01781-f002:**
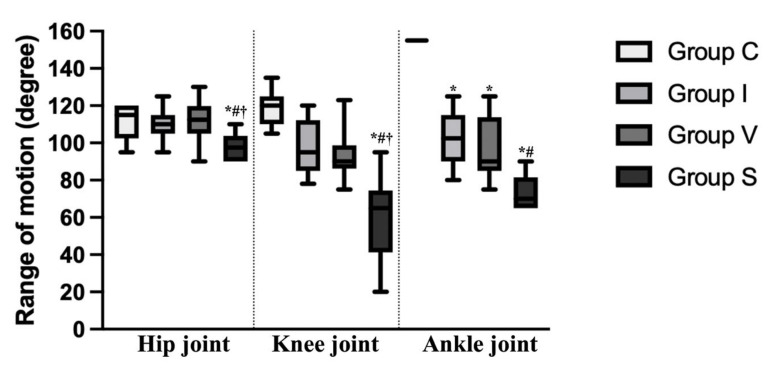
Range of motion of the hip, knee, and ankle joint of the control and immobilization groups on Day 15. The range of motion of each joint was calculated by subtracting the angle of the flexed position from that of the extended position. Data are expressed as median with interquartile ranges. Group C, control group; Group I, cast immobilization group; Group V, Velcro immobilization group; Group S, spiral wire immobilization group. *; Significant difference from Group C within the same joint (*p* < 0.05). #; Significant difference from Group I within the same joint (*p* < 0.05). †; Significant difference from Group V within the same joint (*p* < 0.05).

**Figure 3 biology-11-01781-f003:**
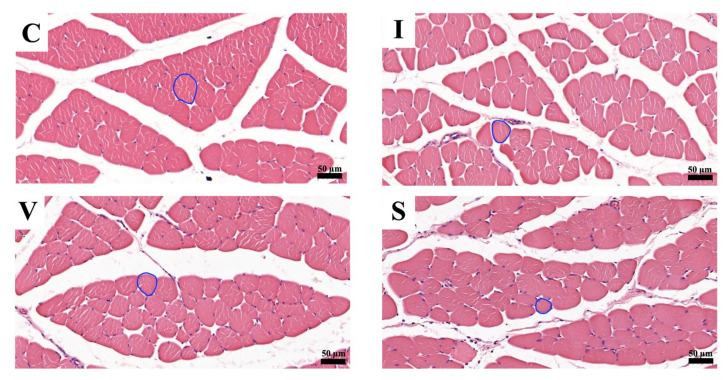
Transverse sections of the quadriceps muscle on Day 15, stained with hematoxylin and eosin at magnification ×200. (**C**) Control group; (**I**) Cast immobilization group; (**V**) Velcro immobilization group; (**S**) Spiral wire immobilization group. The circular line is an example of a round-shaped muscle fiber used for the measurements.

**Figure 4 biology-11-01781-f004:**
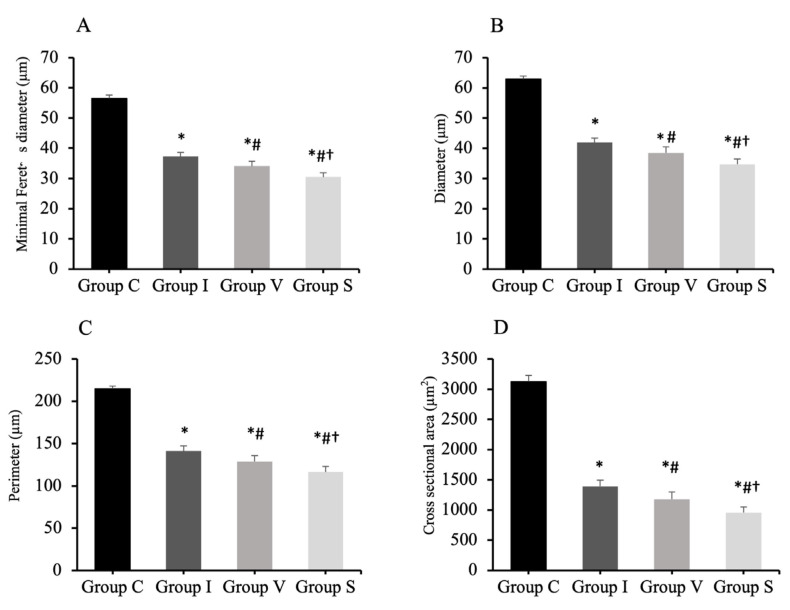
Minimal Feret’s diameter (**A**), diameter (**B**), perimeter (**C**), and cross-sectional area (**D**) of quadriceps muscle fiber analyzed by H&E staining. Group C, control group; Group I, cast immobilization group; Group V, Velcro immobilization group; Group S, spiral wire immobilization group. Values are expressed as mean and standard deviation. *; Significant difference from Group C (*p* < 0.0001). #; Significant difference from Group I (*p* < 0.0001). †; Significant difference from Group V (*p* < 0.0001).

**Figure 5 biology-11-01781-f005:**
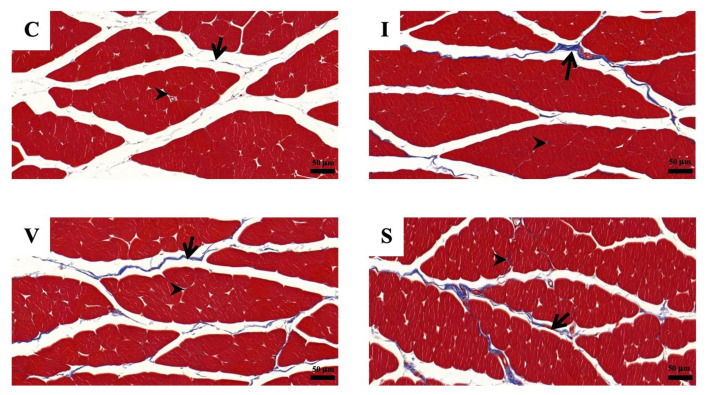
The photomicrographs of Masson’s trichrome staining of quadriceps muscle cross-sections under a light microscope, in Group C (**C**), Group I (**I**), Group V (**V**), and Group S (**S**) on Day 15. (**C**) Control group; (**I**) Cast immobilization group; (**V**) Velcro immobilization group; (**S**) Spiral wire immobilization group. Blue-stained color is collagen, while the red-stained color is muscle fiber. Arrows indicate perimysium, arrowheads indicate endomysium. Magnification at ×200.

**Figure 6 biology-11-01781-f006:**
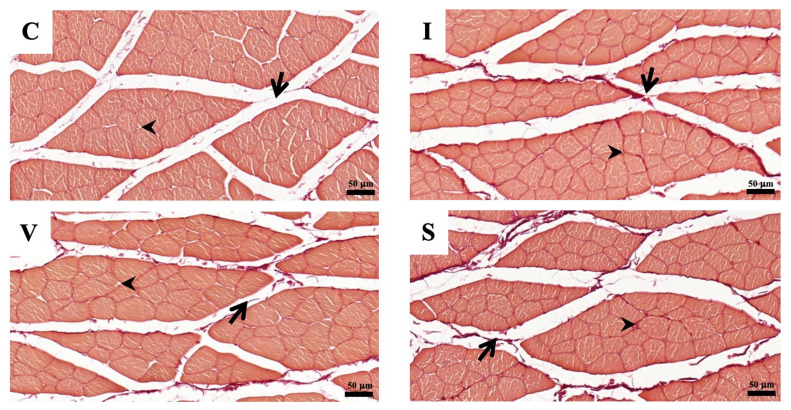
The photomicrographs of Picrosirius red staining of quadriceps muscle cross-sections under a light microscope, in Group C (**C**), Group I (**I**), Group V (**V**), and Group S (**S**) on Day 15. (**C**) Control group; (**I**) Cast immobilization group; (**V**) Velcro immobilization group; (**S**) Spiral wire immobilization group. Dark-red-stained color is collagen, while orange to light-red-counterstained color is muscle fiber. Arrows indicate perimysium, arrowheads indicate endomysium. Magnification at ×200.

**Table 1 biology-11-01781-t001:** Body weight (g) of rats in the control group and the three immobilization groups for 15 days.

Day	Group			
	C (*n* = 5)	I (*n* = 6)	V (*n* = 6)	S (*n* = 6)
1	211.1 ± 7.0	211.9 ± 9.6	216.6 ± 8.4	212.6 ± 7.3
2	221.2 ± 8.0 ^a^	210.8 ± 10.6	210.2 ± 10.2 ^a^	223.1 ± 9.5 ^a^
5	239.0 ± 9.1 ^ab^	207.1 ± 10.5 *	211.2 ± 6.4 *	215.5 ± 6.7 *
8	259.9 ± 14.9 ^abc^	214.4 ± 14.0 *	215.4 ± 6.8 *	222.7 ± 8.8 *
11	279.7 ± 14.3 ^abcd^	213.7 ± 9.2 *	217.2 ± 11.0 *	227.1 ± 11.6 *^c^
14	297.1 ± 15.0 ^abcde^	219.4 ± 15.4 *	220.2 ± 9.9 *	237.4 ± 11.6 *^abc^
15	299.4 ± 14.6 ^abcde^	218.1 ± 18.8 *	222.9 ± 12.4 *	234.5 ± 11.4 *^abc^

Values are expressed as mean ± SD. C; control group; I, cast immobilization group; V, Velcro immobilization group; S, spiral wire immobilization group. *; Significant difference from Group C in the same day (*p* < 0.01). ^a^; Significant difference from Day 1 in the same group (*p* < 0.05). ^b^; Significant difference from Day 2 in the same group (*p* < 0.05). ^c^; Significant difference from Day 5 in the same group (*p* < 0.05). ^d^; Significant difference from Day 8 in the same group (*p* < 0.05). ^e^; Significant difference from Day 11 in the same group (*p* < 0.05).

**Table 2 biology-11-01781-t002:** Food intake (g/day) of rats in the control group and the three immobilization groups for 14 days.

Day	Group			
	C (*n* = 5)	I (*n* = 6)	V (*n* = 6)	S (*n* = 6)
2	19.0 ± 2.2	11.6 ± 5.4	15.2 ± 5.9	17.3 ± 3.5
5	22.2 ± 1.1	17.4 ± 2.7 *	17.9 ± 1.7 *	18.2 ± 1.3 *
8	22.8 ± 1.7	20.1 ± 2.3	19.1 ± 1.3 *	20.4 ± 2.2
11	23.7 ± 0.9	19.6 ± 1.6 *^a^	20.5 ± 3.3	21.2 ± 3.0
14	23.2 ± 0.8 ^a^	19.8 ± 2.6 ^a^	20.4 ± 1.3 *	23.0 ± 2.7 ^a^

Values are expressed as mean ± SD. C, control group; I; cast immobilization group; V, Velcro immobilization group; S, spiral wire immobilization group. *; Significant difference from Group C in the same day (*p* < 0.05). ^a^; Significant difference from Day 2 in the same group (*p* < 0.05).

**Table 3 biology-11-01781-t003:** Bandage application time (minutes) and the frequency of bandage rewinding (times) in the immobilization groups.

Measurements	Group		
	I (*n* = 6)	V (*n* = 6)	S (*n* = 6)
Application time (minutes)	4.6 (4.0–5.5)	7.5 (6.3–8.8)	12.6 * (9.9–13.0)
Frequency of rewinding (times)	1.0 (0.8–1.0)	2.5 ^#^ (1.8–4.3)	1.5 (0.8–2.0)

Values are expressed as median with interquartile ranges. The application time was counted from the bandaging of the left and right sides. The frequency of rewinding was counted from the number of time when rebandage. Bandage was rewinding in both sides in every time. Values are expressed as median with interquartile ranges. I, cast immobilization group; V, Velcro immobilization group; S, spiral wire immobilization group. *; Significant difference from Group I (*p* < 0.001). ^#^; Significant difference from Group I (*p* < 0.05).

**Table 4 biology-11-01781-t004:** Joint angle (°) of rats in the control group and the three immobilization groups on Day 15.

Joint and Position		Group			
		C (*n* = 10)	I (*n* = 12)	V (*n* = 12)	S (*n* = 12)
Hip	Extension	150.0 (147.5–155.0)	157.5 (155.0–165.0)	157.5 (150.0–163.8)	147.5 (145.0–155.0)
	Flexion	41.5 (27.5–51.3)	45.0 (41.3–50.0)	45.0 (40.0–45.0)	55 ^ac^ (45.0–55.0)
Knee	Extension	160.0 (155.0–161.3)	170.0 ^a^ (165.0–173.8)	160.0 (160.0–165.0)	169.0 ^a^ (165.0–170.0)
	Flexion	40.0 (35.0–42.5)	72.5 ^a^ (60.0–80.0)	72.5 ^a^ (61.3–75.0)	97.5 ^abc^ (90.0–133.8)
Ankle	Extension	160.0 (160.0–160.0)	170.0 ^a^ (165.0–170.0)	160.0 (160.0–165.0)	172.5 ^a^ (170.0–175.0)
	Flexion	5.0 (5.0–5.0)	65.0 ^a^ (55.0–80.0)	70.0 ^a^ (55.0–75.0)	95.0 ^abc^ (88.5–107.3)

Values are expressed as median with interquartile ranges. C, control group; I, cast immobilization group; V, Velcro immobilization group; S, spiral wire immobilization group. ^a^; Significant difference from Group C in the same position of the joint (*p* < 0.05). ^b^; Significant difference from Group I in the same position of the joint (*p* < 0.05). ^c^; Significant difference from Group V in the same position of the joint (*p* < 0.05).

**Table 5 biology-11-01781-t005:** Quadriceps muscle weight and quadriceps muscle measurements on Day 15.

Quadriceps Muscle	Group			
	C (*n* = 10)	I (*n* = 12)	V (*n* = 12)	S (*n* = 12)
Muscle weight (g)	2.1 (1.8–2.2)	1.2 (1.1–1.3)	1.3 (1.2–1.3)	1.0 ^abc^ (1.0–1.1)
Muscle weight/Body weight (mg/g)	7.1 (5.7–7.4)	5.5 (5.1–5.8)	5.7 (5.5–5.9)	4.4 ^abc^ (4.2–4.6)
Muscle measurement (mm)				
Width	14.3 (14.1–15.2)	11.6 ^a^ (11.2–11.9)	12.5 ^a^ (11.6–13.4)	11.8 ^a^ (11.0–12.7)
Length	26.0 (24.4–26.9)	23.8 (22.6–24.9)	24.1 (23.6–25.1)	22.8 ^a^ (22.4–23.5)
Height	9.3 (8.2–9.7)	7.3 ^a^ (6.9–7.6)	7.5 ^a^ (6.8–8.3)	7.1 ^a^ (6.1–7.8)

Values are expressed as median with interquartile ranges. C, control group; I, cast immobilization group; V, Velcro immobilization group; S, spiral wire immobilization group. ^a^; Significant difference from Group C (*p* < 0.05). ^b^; Significant difference from Group I (*p* < 0.05). ^c^; Significant difference from Group V (*p* < 0.05).

**Table 6 biology-11-01781-t006:** Collagen percentage in quadriceps muscle of rats that analyzes from Masson’s trichrome staining and Picrosirius red staining on Day 15.

Collagen Amount (%)	Group			
	C (*n* = 10)	I (*n* = 12)	V (*n* = 12)	S (*n* = 12)
**Masson’s trichrome** Collagen percentage	0.5 (0.4–0.7)	1.0* (0.8–1.5)	1.1 * (0.9–1.4)	1.5 * (1.3–1.7)
**Picrosirius red** Collagen percentage	0.9 (0.6–1.2)	2.1 * (1.7–2.4)	2.0 * (1.5–2.5)	2.9 * (2.3–3.3)

The measurement for each rat were averaged over the entire area of transverse quadriceps section at ×400 magnification. Values are expressed as median with interquartile ranges. C, control group; I, cast immobilization group; V, Velcro immobilization group; S, spiral wire immobilization group. *; Significant difference from Group C (*p* < 0.05).

## Data Availability

Not applicable.

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
