# Peer review of "Rat Model of Quadriceps Contracture by Joint Immobilization"

_biology, 2022, doi:10.3390/biology11121781_

Round 1

Reviewer 1 Report

The authors compared 3 methods to immobilize joints in rats. Bilateral hindlimbs were immobilized with either a cast (I), a Velcro hook-and-loop fastener (V), or steel wire (S) with the knee and tarsal joints in extension position for 2 weeks. They found that knee joint range of motion was only decreased by the S method.

My concerns are as follows.

The originality of the study is not clearly highlighted, nor its interest in the context of human pathology (Introduction).

The paper addresses only descriptive data, not mechanism, so the explanation for the differences they observe remains unclear. Why the range of motion is reduced by the S method, for example?

The authors do not sufficiently define terms such as: contracture (line 37), extensibility (line 48), myogenic contracture (line 49), elasticity (line 52).

In my opinion, it is not enough to measure only 100 muscle fibers per muscle (line 190).

It is not necessary to measure the CSA, the diameter and the perimeter (line 357) of the muscle fibers at the same time. The authors can just measure the smallest diameter (min feret), for example.

Why did the autors measure the amount of connective tissue in 2 different ways?

The authors do not mention the functional consequences. It would seem interesting to me to do so, especially in the context of human pathology (Discussion).

Author Response

We greatly appreciate the time and efforts given by the reviewers to make better our manuscript. "Please see the attachment."

Reviewer 2 Report

This manuscript entitled “Rat model of quadriceps contracture by joint immobilization” compared three different types of leg immobilization (casts, Velcro fasteners and steel wires) to evaluate the effects of muscle contracture, and joint movements. The authors checked motion range, muscle length, muscle fiber and collagen ratio in these treatments to establish a reliable animal model to study quadriceps contracture. The authors found that although all three treatments were able to induce muscle atrophic changes and fibrosis in quadriceps, rats with steel wires exhibited significantly low motion range, and less stress (defined by differences between flexion and extension of joint angles) compared to other experimental groups. They therefore concluded that rats with steel wires were the ideal model for studying muscle contracture in human and animals.

Overall, this paper is well-structured and clear-written. The aim of this paper is clear and easy to follow. Muscle contracture induced by long-term leg immobilization is a critical topic in both biomedical and clinic research, and a well-established model will be helpful to the study. However, the link between muscle morphological changes and leg kinematics defects is unclear, and additional experiments are necessary to further validated the proposed model is the best one out of the three models.

Major concern:

1.     The method with steel wires caused a significant low range of motion compared to the other two methods, but the potential mechanism is unclear. It could not be induced by muscle atrophy and fibrosis, as all three experimental groups have similar morphological defects. Further functional studies (such as muscle recordings, EMG) will be required for further address this problem.

2.     Proprioceptors embedded in the joints and tendons monitor leg position and movement and provide mechanical feedback to the motor systems. Therefore, they played an important role in normal locomotion. How do these sensory neurons respond to leg immobilization remains unclear. Are there any morphological changes in these neurons (such as muscle spindle, Golgi tendon organ)?

3.     All the experiments were performed in male rats. Is there any sex-bias for this study?

Minor Concern:

1.     Table 4 and 6 have some numbers mixed with the tables, which need to be fixed.

2.     Yellow tags in Figure 3 are too small to read, rescale is needed.

Author Response

(The authors gave the same response as above.)

Round 2

Reviewer 1 Report

Some problems that I have noted have at least partly been corrected. But the histological results are in my opinion still to be improved. This study remains descriptive. Studying muscle function using electromyography is irrelevant.

Author Response

Thank you very much for your comments. We greatly appreciate the time and efforts given by the reviewers to make better our manuscript. We have changed based on the reviewer’s comments. "Please see the attachment."

Reviewer 2 Report

I'm happy to see that all my concerns and questions have been addressed/corrected in the revised version. I do not have further questions and concerns prior to its publication.

Author Response

Thank you very much for your comments. We greatly appreciate the time and efforts given by the reviewers to make better our manuscript.